# Autoclave treatment of the classical scrapie agent US No. 13-7 and experimental inoculation to susceptible VRQ/ARQ sheep via the oral route results in decreased transmission efficiency

Eric D. Cassmann [ID][☉], Najiba Mammadova[☉], Justin J. Greenlee [ID] *

Virus and Prion Research Unit, National Animal Disease Center, Agricultural Research Service, United States Department of Agriculture, Ames, IA, United States of America

☉ These authors contributed equally to this work.
* justin.greenlee@usda.gov

**Data Availability Statement:** All relevant data are within the manuscript and its Supporting Information files.

## Abstract

Scrapie, a prion disease of sheep, is highly resistant to conventional deactivation. Numerous methods to deactivate scrapie have been tested in laboratory animal models, and adequate autoclave treatment can reduce or remove the infectivity of some classical scrapie strains depending on the heating parameters used. In this study, we autoclaved brain homogenate from a sheep with US scrapie strain 13–7 for 30 minutes at 121˚C. Genetically susceptible VRQ/ARQ sheep were orally inoculated with 3 grams of the autoclaved brain homogenate. For comparison, a second group of sheep was inoculated with a non-autoclaved brain homogenate. Rectal biopsies were used to assess antemortem scrapie disease progression throughout the study. Five out of ten (5/10) sheep that received autoclaved inoculum ultimately developed scrapie after an experimental endpoint of 72 months. These sheep had a mean incubation period of 26.99 months. Two out of five (2/5) positive sheep had detectable PrP$^{Sc}$ in antemortem rectal biopsies, and two (2/5) other sheep had PrP$^{Sc}$ in postmortem rectal tissue. A single sheep (1/5) was positive for scrapie in the CNS, small intestine, and retropharyngeal lymph node but had negative rectal tissue. All of the sheep (10/10) that received non-autoclaved inoculum developed scrapie with a mean incubation period of 20.2 months and had positive rectal biopsies at the earliest timepoint (14.7 months post-inoculation). These results demonstrate that sheep are orally susceptible to US derived classical scrapie strain 13–7 after autoclave treatment at 121˚C for 30 minutes. Differences in incubation periods and time interval to first positive rectal biopsies indicate a partial reduction in infectivity titers for the autoclaved inoculum group.

**Funding:** This research was supported in part by an appointment (N. Mammadova) to the Agricultural Research Service (ARS) Research Participation Program administered by the Oak Ridge Institute for Science and Education (ORISE) through an interagency agreement between the U. S. Department of Energy (DOE) and the U.S. Department of Agriculture (USDA). ORISE is managed by ORAU under DOE contract number DE-SC0014664. All opinions expressed in this paper are the author's and do not necessarily reflect the policies and views of USDA, ARS, DOE, or ORAU/ORISE. This research was funded in its entirety by congressionally appropriated funds to the United States Department of Agriculture, Agricultural Research Service. The funders of the work did not influence study design, data collection and analysis, decision to publish, or the preparation of the manuscript.

## Introduction

Scrapie is a naturally occurring prion disease or transmissible spongiform encephalopathy (TSE) that affects sheep and goats [1]. Other naturally occurring TSEs include bovine spongiform encephalopathy (BSE) in cattle [2,3], chronic wasting disease (CWD) in cervids [4,5], transmissible mink encephalopathy (TME) in mink, and variant Creutzfeldt–Jakob disease (vCJD) in humans [6]. Two main features that distinguish prion diseases from other protein misfolding diseases are their transmissibility and resistance to inactivation by conventional decontamination/sterilization procedures [1,7]. Naturally occurring TSEs of livestock may be transmitted from ingestion of prions shed in bodily fluids (e.g. feces, urine, saliva, placenta tissue) of infected animals, contaminated pastures, and/or decomposing carcasses from dead animals [8–11]. Previous reports have also demonstrated prion infectivity in whole blood or blood fractions of TSE infected animals [10–12]. Therefore, enhanced concern over food safety has prompted numerous studies to investigate potential ways to inactivate prion agents predominantly by biochemical means [13–17] and/or irradiation [18–22] (extensively reviewed in [23] and [24]). A recent study assessed how autoclave treatment affects biochemical stability and infectivity of the atypical scrapie agent Nor98 and the classical scrapie agent PG127 [25]. Transgenic mice (Tg338) were used to demonstrate that autoclave treatment of both scrapie strains significantly reduced infectivity titers and prolonged incubation times after experimental intracerebral inoculation; however, complete inactivation of the prion agents was not achieved [25].

We sought to determine whether standard autoclave treatment of the classical scrapie agent US No. 13–7 and experimental inoculation to susceptible VRQ/ARQ genotype sheep via the oral route would retain prion infectivity. Additionally, we used antemortem rectal biopsies [26,27] to identify positive sheep throughout the experiment. Two groups of genetically susceptible VRQ/ARQ sheep were inoculated with 30 ml of either non-autoclaved or autoclaved scrapie US No. 13–7 via the oral route; serial rectal biopsies were taken from asymptomatic sheep. At the completion of this study, we found that autoclave treatment of the classical scrapie agent and experimental inoculation to susceptible VRQ/ARQ sheep via the oral route resulted in decreased attack rate and significantly increased incubation times compared to sheep that were inoculated with non-autoclaved scrapie. Moreover, autoclave treatment of the classical scrapie agent reduced the likelihood of an antemortem diagnosis by means of rectal biopsy. In this study, we expand on previous reports that investigate the efficacy of various decontamination techniques on different strains of TSEs in an effort to minimize risk of disease transmission.

## Materials and methods

### Ethics statement

The laboratory and animal experiments were conducted in Biosafety Level 2 spaces that were inspected and approved for importing prion agents by the US Department of Agriculture, Animal and Plant Health Inspection Service, Veterinary Services. The studies were done in accordance with the Guide for the Care and Use of Laboratory Animals (Institute of Laboratory Animal Resources, National Academy of Sciences, Washington, DC, USA) and the Guide for the Care and Use of Agricultural Animals in Research and Teaching (Federation of Animal Science Societies, Champaign, IL, USA). The protocols were approved by the Institutional Animal Care and Use Committee at the National Animal Disease Center (protocol number: 3892), which requires species-specific training in animal care for all staff handling animals.

## Animals

This study consisted of twenty VRQ/ARQ sheep that were inoculated at 2 months via the oral route with 30 ml of 10% w/v (3 grams) pooled brain homogenate prepared from whole brains derived from sheep 3441, 3452, 3448, and 3443 from the second serial passage of the US No. 13–7 scrapie isolate in ARQ/ARQ sheep [28]. Brains were homogenized prior to autoclave treatment. Thirty (30) ml of 10% homogenate was added to ten 100 ml glass vials. The glass vials were autoclaved at 15 psi (1 Bar) and 121°C for 30 minutes. Ten of the twenty sheep received autoclaved inoculum. The other 10 sheep were inoculated with non-autoclaved homogenate. The procedure for oral inoculation of lambs has been described previously [29]. All inoculated sheep were housed in biosafety level 2 facilities following exposure to scrapie. The sheep were fed pelleted growth ration and alfalfa hay, and clean water was available *ad libitum*. Non-inoculated control sheep (n = 4) were kept with the scrapie-free flock at the NADC. Animals were observed daily for the development of clinical signs of neurologic disease and were euthanized at the onset of unequivocal clinical signs of disease. The method of euthanasia was intravenous administration of sodium pentobarbital as per label directions or as directed by an animal resources attending veterinarian. Clinical signs of disease included abnormalities in gate and/or stance, and ataxia. Rectoanal mucosal biopsies were collected from living sheep at three timepoints: 14.7, 16.1, and 23.5 months post inoculation. The biopsy procedure was performed with a rectal speculum and lubricant containing 0.2% lidocaine for analgesia. The rectoanal mucosa was visualized, the mucosa was elevated with forceps, and a 1 cm diameter piece of tissue was excised with scissors. Incubation period is reported here as the time from inoculation to the time when unequivocal signs of clinical disease are present. Survival curves were created with Prism 6 for Windows (Graph Pad Software, Version 6.01); both the logrank and Gehan-Breslow-Wilcoxon tests [30] were used to test the null hypothesis that survival curves were identical between treatment groups with a significance level of alpha set at 0.05. The difference between the mean incubation periods was analyzed with a two-tailed unpaired t-test (alpha = 0.05).

At necropsy, duplicate tissue samples were collected and either frozen or stored in 10% buffered neutral formalin. Specifically, tissues were collected comprising representative sections of the brain, spinal cord, retinas, pituitary, trigeminal ganglia, sciatic nerve, third eyelids, tonsils (palatine and pharyngeal), lymph nodes (retropharyngeal, prescapular and popliteal), spleen, esophagus, forestomaches, intestines, rectal mucosa, thymus, liver, kidney, urinary bladder, pancreas, salivary gland, thyroid gland, adrenal gland, trachea, lung, turbinates, heart, tongue, masseter muscle, diaphragm, triceps brachii, biceps femoris, and psoas major.

## Immunohistochemistry, enzyme immunoassay (EIA), and western blot analysis

For detection of PrP$^{Sc}$, slides were stained by an automated immunohistochemistry method using a cocktail of primary antibodies F99/F96.7.1 and F89/160.1.5 as described previously [31,32]. Briefly, paraffin-embedded sections (4 μm) were rehydrated using xylene, followed by a decreasing ethanol concentration gradient (100%, 90% 70%), and a final wash with diH$_2$O. Heat-mediated antigen retrieval was performed using citrate buffer (DAKO Target Retrieval Solution, DAKO Corp., Carpinteria, CA, USA) in an autoclave for 30 min. Slides were then stained with an indirect, biotin free staining system containing an alkaline phosphatase labeled secondary antibody (*ultra*View Universal Alkaline Phosphatase Red Detection Kit, Ventana Medical Systems, Inc., Tucson, AZ) designed for an automated immunostainer (NexES IHC module, Ventana Medical Systems). Slides were counterstained with Gill's hematoxylin and bluing agent (Ventana Medical Systems) and then cover slipped. Images were captured using a

Nikon Eclipse 50i microscope with a Nikon DSfi-2 camera and a DS-L3 controller (Nikon Instruments Inc., Melville, NY).

A commercially available enzyme immunoassay (HerdChek®, IDEXX Laboratories Inc., Westbrook, ME) was used to screen for the presence of PrP$^{Sc}$ in brainstem at the level of the obex and the retropharyngeal lymph node (RPLN). Assays were conducted according to kit instructions except that the samples were prepared as a 20% (w/v) tissue homogenate. Cut-off numbers were determined with a negative control per kit instructions; values greater than the mean optical density (O.D.) of negative controls +0.180 were considered positive for the purposes of screening samples.

For western blot analysis, approximately 0.5–1 milligram of brainstem was analyzed as described previously, with minor modifications [33]. Samples were homogenized at 4˚C in PBS and digested with proteinase K (PK) for 1 hour @ 37˚C. PK-digestion was stopped using Pefabloc (Roche, Indianapolis, IN) to a final concentration of 1 mg/ml. Samples were acetone precipitated, resuspended with 1x LDS loading buffer, and loaded into a pre-cast sodium dodecyl sulfate (SDS)-12% polyacrylamide gel electrophoresis (PAGE) gel. SDS-PAGE was performed as described by the manufacturer and the proteins were transferred from the gel to a PVDF membrane with transfer buffer at 25 V for 60 minutes. The membranes were blocked with 3% BSA in TBS-T (Tris-Buffered Saline + 0.05% Tween-20) and incubated with monoclonal antibody P4 at 0.1 µg/mL for 1 hour at room temperature. Biotinylated sheep anti-mouse IgG secondary antibody (GE Healthcare, Buckinghamshire, UK) at 0.1 µg/mL, streptavidin-horseradish peroxidase (HRP) conjugate (GE Healthcare, Buckinghamshire, UK) at 0.1 µg/mL, and a chemiluminescent detection system (ECL Plus detection system, GE Healthcare, Buckinghamshire, UK) were used for western blotting in conjunction with a digital imager (GBOX, Synoptics).

## Results

To determine if the scrapie agent US No. 13–7 retained sufficient levels of prion infectivity to cause disease after heat-treatment, genetically susceptible VRQ/ARQ sheep were orally inoculated with 3 grams of either non-autoclaved or autoclaved scrapie US No. 13–7. Furthermore, to track disease progression in asymptomatic sheep, three serial biopsies of rectoanal mucosa-associated lymphoid tissue (RAMALT) were taken over the course of the observation period to test for PrP$^{Sc}$ accumulation by immunohistochemistry. All ten animals (10/10) that were inoculated with non-autoclaved scrapie were determined scrapie positive based on accumulation of prion protein by immunohistochemistry, western blot, and/or enzyme immunoassay (EIA) in CNS and non-CNS tissues (Table 1). All ten animals showed clinical signs of scrapie and had a mean incubation period of 20.21 ± 0.81 (mean ± SEM) months post-inoculation (MPI) (Table 1). Additionally, these ten animals had two positive antemortem rectal biopsies before they presented with neurologic clinical signs and were euthanized (Table 2).

Of the ten animals that were inoculated with autoclaved scrapie, five (50%) were determined scrapie positive and had a significantly longer mean incubation period (p<0.0004) of 26.99 ± 1.23 (mean ± SEM) months compared to 20.21 ± 0.81 months in sheep that received non-autoclaved inoculum (Fig 1). There was a significant difference in the survival curves between the experimental groups using both the logrank and Gehan-Breslow-Wilcoxon tests (p<0.0001 and p<0.0001, respectively). Immunoreactivity directed against PrP$^{Sc}$ was observed in both lymphoid and CNS tissue of positive sheep (Fig 2). Animal 270 had PrP$^{Sc}$ in the CNS, retropharyngeal lymph nodes, gut-associated lymphoid tissue of the small intestines and the enteric nervous system; however, no PrP$^{Sc}$ was detected in the tonsils (pharyngeal and palatine), spleen, or rectoanal mucosa-associated lymphoid tissue. Of the five animals that were

**Table 1. Immunohistochemistry of CNS and non-CNS tissues.**

| | | | | Immunohistochemistry for PrP^Sc | | | | | | | | |
| | | | | CNS | | Lymphoid | | | | | | |
| Inoculum | ID # | EIA Obex | MPI | Obex | Retina | RPLN | Phgl. Tonsil | Pal. Tonsil | Spleen | RAMALT | GALT, SI | ENS, GI |
|---|---|---|---|---|---|---|---|---|---|---|---|---|
| Non- Autoclaved scrapie US No. 13–7 | 242 | + | 18.00 | + | + | + | + | + | + | + | + | + |
| | 245 | + | 22.15 | + | + | + | + | + | + | + | + | + |
| | 246 | + | 22.15 | + | + | + | + | + | + | N/A | + | + |
| | 254 | + | 14.41 | + | - | N/A | + | N/A | N/A | IS | N/A | N/A |
| | 269 | + | 21.96 | + | + | + | + | + | + | + | N/A | N/A |
| | 275 | + | 20.37 | + | + | + | + | N/A | + | + | + | + |
| | 276 | + | 22.12 | + | + | + | + | + | + | IS | + | + |
| | 304 | + | 22.12 | + | + | + | + | + | + | + | + | + |
| | 305 | + | 18.73 | + | + | + | + | + | + | + | + | + |
| | 312 | + | 20.11 | + | + | N/A | + | + | + | + | + | + |
| Autoclaved scrapie US No. 13–7 | 231 | - | 71.51 | - | - | - | N/A | - | - | - | - | - |
| | 235 | - | 15.89 | - | - | - | - | - | - | IS | - | - |
| | 237 | + | 22.85 | + | + | + | N/A | + | + | + | + | + |
| | 243 | - | 71.54 | - | - | - | - | - | - | - | - | - |
| | 244 | - | 71.54 | - | - | - | - | - | - | - | - | - |
| | 255 | + | 26.04 | + | + | + | + | + | + | + | + | + |
| | 270 | + | 30.23 | + | + | + | - | - | - | - | + | + |
| | 288 | + | 27.92 | + | + | + | + | + | + | + | + | + |
| | 289 | + | 27.92 | + | + | + | + | + | + | + | + | + |
| | 323 | - | 71.74 | - | - | - | - | - | - | - | - | - |

Summary table of scrapie transmission results in sheep inoculated via the oral route with either non-autoclaved or autoclaved scrapie agent US No. 13–7. Abbreviations: EIA, enzyme immunoassay; MPI, months post inoculation; RPLN; retropharyngeal lymph node; Phgl. Tonsil, pharyngeal tonsil; Pal. Tonsil; palatine tonsil; RAMALT, rectoanal mucosa-associated lymphoid tissue; GALT SI, gut-associated lymphoid tissue of the small intestines; ENS, enteric nervous system; N/A, tissue not available for examination; IS, insufficient sample.

PrP^Sc positive at the end of study, only two animals (288 and 289) had a positive antemortem rectal biopsies. These biopsies were taken approximately 4.5 months before they developed clinical signs of disease and were euthanized.

The molecular profile of PrP^Sc from brainstem homogenates was analyzed by western blot, to compare US No. 13–7 scrapie isolate from an ARQ/ARQ sheep with a sheep orally inoculated with autoclaved scrapie, non-autoclaved scrapie, and a sheep orally inoculated with autoclaved scrapie that was scrapie negative (Fig 3). Western blot analysis revealed a similar banding pattern between all groups.

## Discussion

These data show that sheep were susceptible to classical scrapie agent 13–7 after autoclave treatment at 121˚C for 30 minutes. These sheep had a prolonged incubation period compared to sheep that received non-autoclaved inoculum. Two groups of VRQ/ARQ genotype sheep were orally inoculated with 3 grams of either non-autoclaved or autoclaved scrapie US No. 13–7. We used antemortem rectal biopsies to identify positive sheep throughout the experiment. At the completion of the study, five of the ten animals (50%) that were inoculated with autoclaved scrapie were determined scrapie positive and had a significantly longer mean incubation period (~7 months) compared to sheep that were inoculated with non-autoclaved scrapie. Moreover, of these five animals that were PrP^Sc positive at the end of study, two animals (40%)

**Table 2. Immunohistochemistry of antemortem serial rectal biopsies.**

| Inoculum | Animal ID | Incubation Time (MPI) | Rectal Biopsies | | | | Final Scrapie Status |
|---|---|---|---|---|---|---|---|
| | | | 1 | 2 | 3 | PM | |
| Non- Autoclaved scrapie US No. 13–7 | 242 | 18.00 | + | + | XX | + | + |
| | 245 | 22.15 | + | + | XX | + | + |
| | 246 | 22.15 | + | + | XX | NA | + |
| | 254 | 14.41 | XX | XX | XX | IS | + |
| | 269 | 21.96 | + | + | XX | + | + |
| | 275 | 20.37 | + | + | XX | + | + |
| | 276 | 22.12 | + | + | XX | IS | + |
| | 304 | 22.12 | + | + | XX | + | + |
| | 305 | 18.73 | + | + | XX | + | + |
| | 312 | 20.11 | + | + | XX | + | + |
| Autoclaved scrapie US No. 13–7 | 231 | 71.51 | - | - | - | - | - |
| | 235 | 15.89 | - | NT | XX | IS | - |
| | *237 | 22.85 | - | - | XX | + | + |
| | 243 | 71.54 | - | - | - | - | - |
| | 244 | 71.54 | - | - | - | - | - |
| | *255 | 26.04 | - | - | - | + | + |
| | *270 | 30.23 | - | - | - | - | + |
| | †288 | 27.92 | - | - | + | + | + |
| | †289 | 27.92 | - | - | + | + | + |
| | 323 | 71.74 | - | - | - | - | - |

Summary table of results obtained from antemortem rectal biopsies tested for PrP$^{Sc}$ by immunohistochemistry. Rectal biopsies of mucosal associated lymphoid tissue were taken at timepoints 1, 2, and 3 that corresponded to 14.7, 16.1, and 23.5 months post-inoculation (MPI), respectively. Animals 237, 255, and 270 were determined scrapie positive, but had negative antemortem rectal biopsies (*). Sheep 270 had scrapie, but it was rectal tissue negative at the time of post-mortem examination. Sheep 288 and 289 were determined scrapie positive and had one positive antemortem rectal biopsy (†). Sheep died before the antemortem rectal biopsy was taken (XX). Abbreviations: PM, post-mortem; NT, not tested; MPI, months post inoculation. IS, insufficient sample. NA, not available.

had only one positive antemortem RAMALT biopsy detected at ~83% of their incubation period, or 4.5 months before the onset of clinical signs.

This study expands on early work that investigated the altered infectivity of the scrapie agent; numerous methods for deactivation include biochemical and ionizing radiation [14,16,18–20,34]. Under certain conditions, it has been reported that heat-based deactivation can stabilize TSE agents and make them more resistant to inactivation [35], but a combination of biochemical and physical deactivation methods increases deactivation [36–38]. Other research shows that simple heat deactivation of scrapie is dependent on temperature and duration [39]. At temperatures of 121°C for 90 minutes, strains Sc237 and 263K were not completely deactivated [36,40]. Raising the temperature and duration of autoclave treatment can be more effective; for example, autoclaving at 132°C for 90 minutes completely reduced detectable infectivity in strain 263K. Notably, there is a reported difference in the ability of autoclaving to inactivate different scrapie strains [41,42].

The present research is unique for two reasons. First, we investigate the bio-relevance of scrapie deactivation in the natural host species. The experiments discussed above utilized wild type mice to estimate changes in infectivity. To date, the relevance of various inactivation methods on infectivity in sheep is unknown. Second, this is the first investigation of retained infectivity after autoclave treatment using US classical scrapie strain 13–7. While sheep have not been used to evaluate residual infectivity after autoclave decontamination, Spiropoulos et.

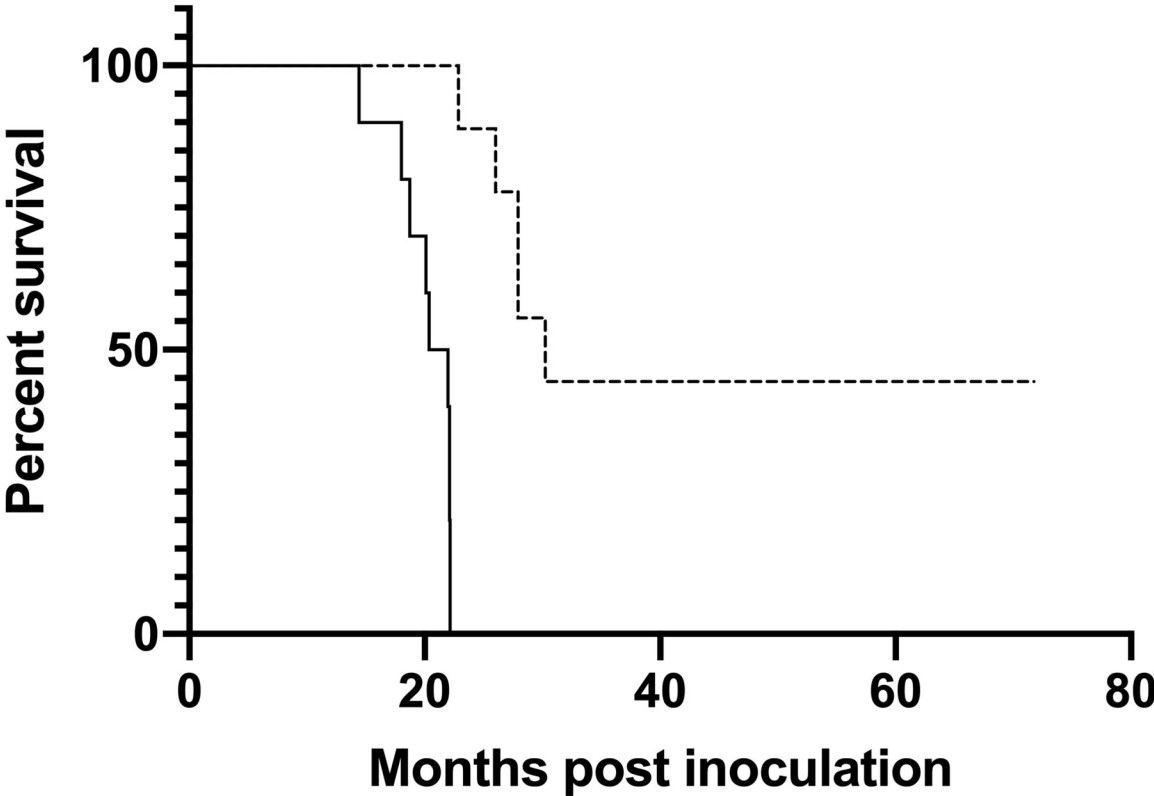

**Fig 1. Death events in sheep inoculated via the oral route with either non-autoclaved or autoclaved scrapie agent US No. 13–7.**
Percent survival graph showing a significant difference (p<0.0001, Gehan-Breslow-Wilcoxon test) between survival curves.

al. used mice expressing ovine $PRNP_{VRQ}$ to assess infectivity of classical scrapie strain PG127 after autoclaving at 133˚C, 3 Bar (43.5 psi), for 20 minutes [43]. Our observation of a prolonged incubation period in sheep paralleled their findings in transgenic mice. Of course, direct comparisons are difficult owing to transgenic mice versus sheep, different autoclave parameters, and a different classical scrapie strain.

The autoclave parameters used in the present study were below the current USDA APHIS recommendations outlined in the National Scrapie Eradication Program: Scrapie Program Standards Vol 1, Appendix E [44]. The standards direct autoclaving at 136˚C for 1 hour, and decontamination is enhanced by pretreatment with sodium hydroxide or sodium hypochlorite. We evaluated the infectivity of a classical scrapie strain after autoclaving at 121˚C for 30 minutes. These cycle parameters were chosen because they are commonly used for microbial deactivation. At this temperature and duration, US scrapie strain 13–7 was still infectious to sheep after oral inoculation. This finding emphasizes the need to avoid substandard decontamination temperatures when autoclaving materials that could be exposed to scrapie prions.

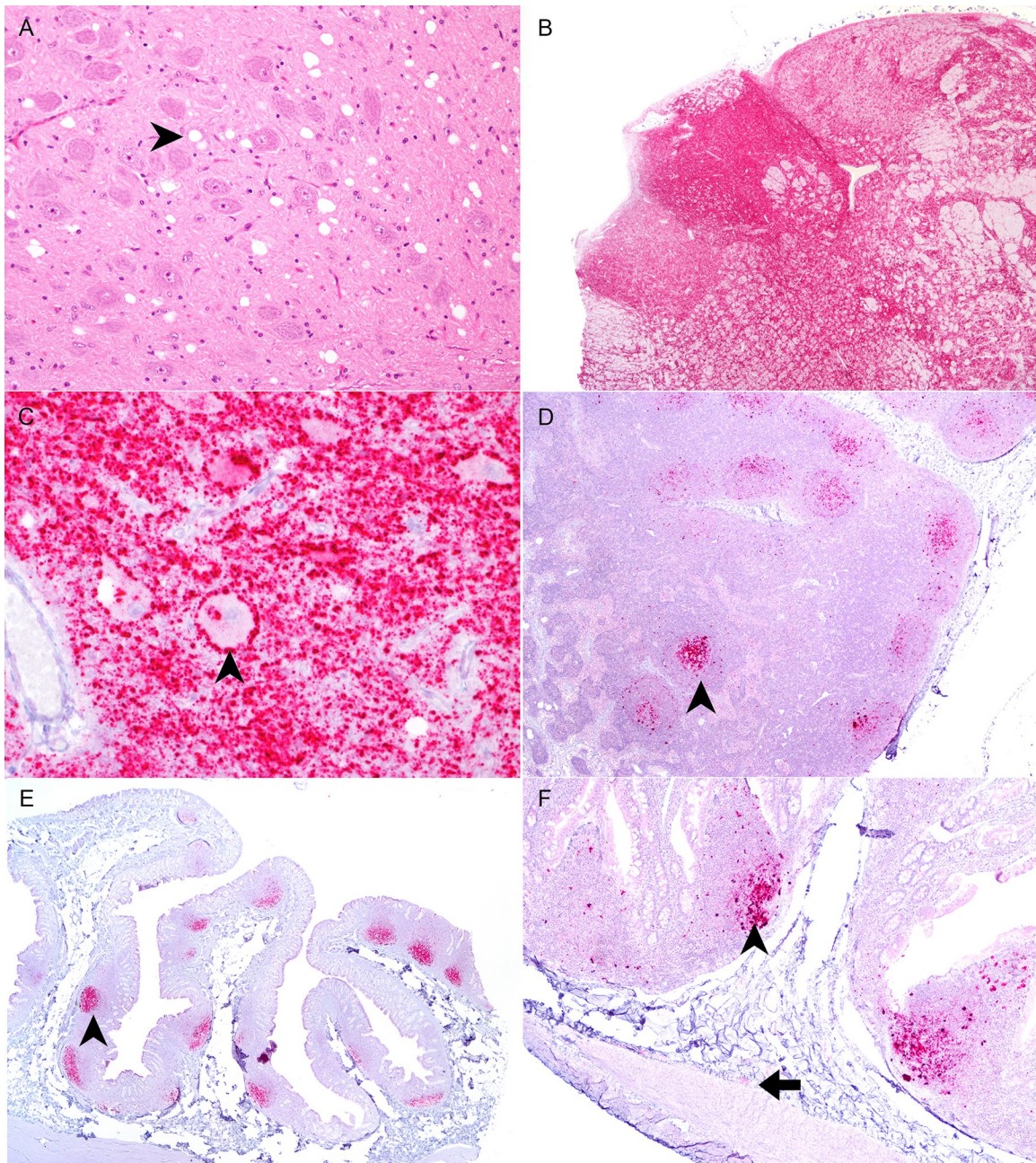

**Fig 2. Spongiform change and immunohistochemistry for PrP<sup>Sc</sup> in tissues from sheep orally inoculated with autoclaved scrapie agent US No. 13–7.** (A) Spongiform encephalopathy in sheep #255 that received autoclaved inoculum in the brainstem at the level of the obex (hematoxylin and eosin). An intraneuronal vacuole is indicated by the arrowhead. (B) Immunoreactivity for PrP<sup>Sc</sup> (red) in the brainstem at the level of the obex in sheep #255. (C) There is abundant coarse granular particulate in the neuropil. Perineuronal and intraneuronal PrP<sup>Sc</sup> is present (arrowhead). (D) There is positive immunolabeling (red) in a follicle (arrowhead) of the retropharyngeal lymph node of sheep #237. (E) Follicles in the rectoanal mucosa-associated lymphoid tissue are immunopositive for PrP<sup>Sc</sup> (arrowhead). (F) PrP<sup>Sc</sup> is present in the small intestinal mucosa-associated lymphoid tissue (arrow) and neurons of the enteric nervous system (arrowhead).

We demonstrated that VRQ/ARQ genotype sheep are orally susceptible to 3 grams of brain homogenate containing classical scrapie prions that was autoclaved for 30 minutes at 121˚C. At the experimental endpoint of 72 months post-inoculation, 50% (5/10) of the sheep that

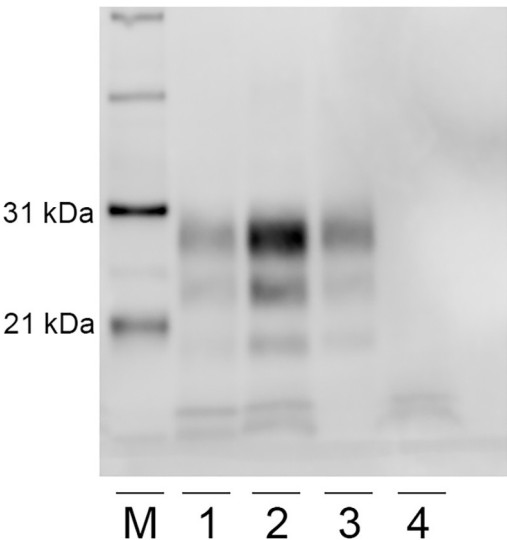

**Fig 3. Western blot migration patterns of ovine scrapie.** Proteinase K-digestion of brain homogenates from sheep orally inoculated with scrapie reveals three immunoreactive bands that represent three glycoforms (monoclonal antibody P4). Western blot analysis of PrP$^{Sc}$ reveals similar band patterns of brain homogenate prepared from a whole brain derived from a sheep from the second passage of the US No. 13–7 scrapie isolate in ARQ/ARQ sheep (lane 1), scrapie positive sheep (288) orally inoculated with autoclaved scrapie US No. 13–7 (lane 2), and scrapie positive sheep (242) orally inoculated with non-autoclaved scrapie US No. 13–7 (lane 3). Lane 4 represents a scrapie negative sheep (243) inoculated with autoclaved scrapie US No. 13–7. Two low molecular weight non-specific bands are present in lanes 1, 2, and 4. M: molecular weight marker.

received autoclaved inoculum had succumbed to scrapie. Positive sheep had a prolonged incubation period compared to the non-autoclave inoculum group. Only 2/5 scrapie positive sheep had positive antemortem RAMALT biopsies from 23.5 months post-inoculation. In contrast, sheep that received non-autoclaved inoculum had positive RAMALT biopsies at the first time-point, 14.7 months post-inoculation. Autoclave treatment resulted in a prolonged interval until the first positive rectal biopsy. This is presumably due to a lower infectious titer in the autoclaved inoculum. In summary, autoclave treatment of brain tissue from a symptomatic scrapie affected sheep was not sufficient to completely abate transmission under our experimental conditions.

## Supporting information

**S1 Fig. Raw image for western blot from Fig 3.**
(PDF)

## Acknowledgments

The authors wish to thank Dr. Robert Kunkle (ret.) and Dennis Orcutt (ret.) for their initial contributions to this work; additionally, we thank Rylie Frese, Kevin Hassall, Joe Lesan, Leisa Mandell, and Trudy Tatum for technical support.

## Author Contributions

**Conceptualization:** Justin J. Greenlee.

**Data curation:** Eric D. Cassmann, Najiba Mammadova.

**Formal analysis:** Eric D. Cassmann, Najiba Mammadova, Justin J. Greenlee.

**Funding acquisition:** Justin J. Greenlee.

**Investigation:** Eric D. Cassmann, Najiba Mammadova, Justin J. Greenlee.

**Methodology:** Justin J. Greenlee.

**Project administration:** Justin J. Greenlee.

**Resources:** Justin J. Greenlee.

**Supervision:** Justin J. Greenlee.

**Writing – original draft:** Eric D. Cassmann, Najiba Mammadova.

**Writing – review & editing:** Eric D. Cassmann, Najiba Mammadova, Justin J. Greenlee.

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
