## [Decision Letter · Decision Letter 0]

5 Oct 2020

PONE-D-20-23931

Autoclave treatment of the classical scrapie agent US No. 13-7 and experimental inoculation to susceptible VRQ/ARQ sheep via the oral route results in decreased transmission efficiency

PLOS ONE

Dear Dr. Greenlee,

Thank you for submitting your manuscript to PLOS ONE. I apologize for the prolonged review process; it was difficult to collect reviews in these crazy times. After careful consideration, we feel that it has merit but does not fully meet PLOS ONE’s publication criteria as it currently stands. Therefore, we invite you to submit a revised version of the manuscript that addresses the points raised during the review process.

As listed below, the reviewers expressed some concerns and suggested ways to improve your manuscript. Among the key concerns that I would ask you to consider are insufficient clarity as to 1) why you chose those particular autoclaving conditions for your study, and 2) the strength of your conclusions in the absence of more direct comparisons of your treatment conditions to other conditions, or, your chosen scrapie strain to other strains. One reviewer felt strongly enough about this to recommend rejection. Also there is the issue of the staining in Fig 2A as well as other issues that are worth serious consideration.

We look forward to receiving your revised manuscript.

Kind regards,

Byron Caughey

Academic Editor

PLOS ONE

Journal Requirements:

2. To comply with PLOS ONE submissions requirements, in your Methods section, please provide additional information on the animal research and ensure you have included details on (1) methods of sacrifice, (2) methods of anesthesia and/or analgesia, and (3) efforts to alleviate suffering.

3.. PLOS ONE now requires that authors provide the original uncropped and unadjusted images underlying all blot or gel results reported in a submission’s figures or Supporting Information files. This policy and the journal’s other requirements for blot/gel reporting and figure preparation are described in detail at https://journals.plos.org/plosone/s/figures#loc-blot-and-gel-reporting-requirements and https://journals.plos.org/plosone/s/figures#loc-preparing-figures-from-image-files. When you submit your revised manuscript, please ensure that your figures adhere fully to these guidelines and provide the original underlying images for all blot or gel data reported in your submission. See the following link for instructions on providing the original image data: https://journals.plos.org/plosone/s/figures#loc-original-images-for-blots-and-gels.

Reviewers' comments:

Reviewer's Responses to Questions

**Comments to the Author**

1. Is the manuscript technically sound, and do the data support the conclusions?

Reviewer #1: Yes

Reviewer #2: No

2. Has the statistical analysis been performed appropriately and rigorously? 

Reviewer #1: Yes

Reviewer #2: Yes

3. Have the authors made all data underlying the findings in their manuscript fully available?

Reviewer #1: Yes

Reviewer #2: Yes

4. Is the manuscript presented in an intelligible fashion and written in standard English?

Reviewer #1: Yes

Reviewer #2: Yes

5. Review Comments to the Author

Reviewer #1: Dear Authors,

Your manuscript "Autoclave treatment of the classical scrapie agent US No. 13-7 and experimental

inoculation to susceptible VRQ/ARQ sheep via the oral route results in decreased

transmission efficiency" was clearly written and the data/conclusion was discussed in a fare and meaningful manner. The experimental design and read-out were quite simple.

A few comments

1. It would be helpful to include the reasoning on why the specific autoclave parameters for this study were selected (place in intro).

2. The text describes the failure to detect PrP in RAMALT biopsies collected from sc+ mice in the group infected with autoclaved tissue. This is interesting, especially since many of the sc+ sheep were still negative even at the time of euthanasia. Can you speculate a reason why you think this is occurring? Were the strain properties/tissue tropisms altered by autoclaving? Is there any precedence for a change in tissue distribution like this following autoclaving or inactivation by other means? ( line 213).

3. Please provide more detail on how the brain/s was autoclaved. How much mass? Containment? Was it whole brain or already homogenized.

4. Figure 1 would depict the data more accurately as a staircase survival curve. As shown, the percentage of sheep alive drops immediately below 100% at day 1. Since the incubation periods are long, and the N values somewhat small in one group, using a staircase style would be preferred.

Reviewer #2: In the current study, the authors assessed the effect of autoclaving at 121°C for 30 minutes on the infectivity of classical scrapie, the US scrapie strain 13-7. Autoclave treatment prolonged the incubation periods on VRQ/ARQ sheep when orally inoculated with the inoculum. It is somewhat hard to understand the goal of the study. If the authors focus on the effect of autoclave treatment on the infectivity, various condition for autoclave should have been set. If the authors focus on the prion strain, different strains should have been compared. From Fig. 2A, there may be a technical issue of PrPSc detection by IHC on the brain tissue. The immune reactivity is too homogenous.

6. PLOS authors have the option to publish the peer review history of their article (what does this mean?). If published, this will include your full peer review and any attached files.

Reviewer #1: No

Reviewer #2: No

---

## [Author Response · Author response to Decision Letter 0]

12 Nov 2020

Journal Requirements:

Response: The files were renamed.

2. To comply with PLOS ONE submissions requirements, in your Methods section, please provide additional information on the animal research and ensure you have included details on (1) methods of sacrifice, (2) methods of anesthesia and/or analgesia, and (3) efforts to alleviate suffering.

 Response: This information was added to the methods (lines 88-89 and 96). 

3.. PLOS ONE now requires that authors provide the original uncropped and unadjusted images underlying all blot or gel results reported in a submission’s figures or Supporting Information files. This policy and the journal’s other requirements for blot/gel reporting and figure preparation are described in detail at https://journals.plos.org/plosone/s/figures#loc-blot-and-gel-reporting-requirements and https://journals.plos.org/plosone/s/figures#loc-preparing-figures-from-image-files. When you submit your revised manuscript, please ensure that your figures adhere fully to these guidelines and provide the original underlying images for all blot or gel data reported in your submission. See the following link for instructions on providing the original image data: https://journals.plos.org/plosone/s/figures#loc-original-images-for-blots-and-gels.

Respose: The uncropped image is uploaded and available for review as supporting information.

Response: Captions have been added. 

Reviewers' comments:

Reviewer's Responses to Questions

Comments to the Author

1. Is the manuscript technically sound, and do the data support the conclusions?

Reviewer #1: Yes

Reviewer #2: No

2. Has the statistical analysis been performed appropriately and rigorously? 

Reviewer #1: Yes

Reviewer #2: Yes

3. Have the authors made all data underlying the findings in their manuscript fully available?

Reviewer #1: Yes

Reviewer #2: Yes

4. Is the manuscript presented in an intelligible fashion and written in standard English?

Reviewer #1: Yes

Reviewer #2: Yes

5. Review Comments to the Author

Reviewer #1: Dear Authors,

Your manuscript "Autoclave treatment of the classical scrapie agent US No. 13-7 and experimental inoculation to susceptible VRQ/ARQ sheep via the oral route results in decreased transmission efficiency" was clearly written and the data/conclusion was discussed in a fare and meaningful manner. The experimental design and read-out were quite simple.

A few comments

1. It would be helpful to include the reasoning on why the specific autoclave parameters for this study were selected (place in intro).

Response: Thank you for the suggestion. The autoclave parameters were selected because they are standard operating settings. This is different than the APHIS recommended parameters (as discussed in the discussion). Per your suggestion, we clarified in the introduction that the autoclave treatment selected was “standard”. 

2. The text describes the failure to detect PrP in RAMALT biopsies collected from sc+ mice in the group infected with autoclaved tissue. This is interesting, especially since many of the sc+ sheep were still negative even at the time of euthanasia. Can you speculate a reason why you think this is occurring? Were the strain properties/tissue tropisms altered by autoclaving? Is there any precedence for a change in tissue distribution like this following autoclaving or inactivation by other means? (line 213).

Response: Only 3 sheep had negative antemortem rectal biopsies and a positive final scrapie status (237 and 255 and 270). There were several months (6, 3, and 7 respectively) between the last antemortem rectal biopsies and postmortem analyses on these sheep. We suspect that lower titers in the inocula led to longer incubation periods and slower lymphoid distribution. It’s also possible that IHC simply missed positive follicles (false negative). A difference in lymphotropism is not likely given the widespread (in RPLN, tonsils, spleen and GALT) lymphoid distribution in 4/5 positive sheep. Even sheep 270 without splenic and tonsillar PrPSc had PrPSc in the retropharyngeal lymph node and the small intestinal GALT.

3. Please provide more detail on how the brain/s was autoclaved. How much mass? Containment? Was it whole brain or already homogenized.

Response: We have added this information to the materials and methods (lines 80-82).

4. Figure 1 would depict the data more accurately as a staircase survival curve. As shown, the percentage of sheep alive drops immediately below 100% at day 1. Since the incubation periods are long, and the N values somewhat small in one group, using a staircase style would be preferred.

Response: We made changes to Figure 1 and redesigned the survival curve. This led to us ascertaining that 5 sheep in the autoclave scrapie group were not included in the first curve. They were added and censored appropriately for surviving to the experimental endpoint. This changed the p-values for our analyses; those results were updated in the manuscript as well. 

Reviewer #2: In the current study, the authors assessed the effect of autoclaving at 121°C for 30 minutes on the infectivity of classical scrapie, the US scrapie strain 13-7. Autoclave treatment prolonged the incubation periods on VRQ/ARQ sheep when orally inoculated with the inoculum. It is somewhat hard to understand the goal of the study. If the authors focus on the effect of autoclave treatment on the infectivity, various condition for autoclave should have been set. If the authors focus on the prion strain, different strains should have been compared. From Fig. 2A, there may be a technical issue of PrPSc detection by IHC on the brain tissue. The immune reactivity is too homogenous.

Response: We appreciate the suggestions for ways to improve this study. We conducted a simple study in order to answer a simple question. That question was would standard autoclaving alone remove infectivity of classical scrapie in the native host? The reviewer mentioned that we discussed “infectivity” and “strain differences” in our paper. These topics became part of the manuscript’s discussion after evaluating the results of our short experiment. In an attempt at writing a thorough discussion, we mentioned that previous research has shown differences in infectivity between strains after decontamination methods. It’s prudent to mention variables that could possible obscure results between different studies. 

In the present experiment, we tested whether or not sheep were orally susceptible to a US classical scrapie strain after standard autoclave parameters. Neither sheep, the native host, nor this strain of scrapie have ever been used in such an experiment before. We appreciate the suggestion to make extra comparisons, but incorporating extra sheep into such a study aimed at answering a simple question would add sizeable expense and duration. The use of the native host species has its benefits, but cost and duration often preclude large study designs testing multiple variables. 

A new figure 2 was made that demonstrates the immunoreactivity more clearly. The amount of PrPSc staining in the neuropil observed at low magnification seems homogenous, but at higher magnification (plate C) individual granules and coarse particulate are observed. These staining characteristics are typical of sheep with clinical disease and it is not consistent with technical staining issues. 

6. PLOS authors have the option to publish the peer review history of their article (what does this mean?). If published, this will include your full peer review and any attached files.

Do you want your identity to be public for this peer review? For information about this choice, including consent withdrawal, please see our Privacy Policy.

Reviewer #1: No

Reviewer #2: No

---

## [Editor Report · Decision Letter 1]

13 Nov 2020

Autoclave treatment of the classical scrapie agent US No. 13-7 and experimental inoculation to susceptible VRQ/ARQ sheep via the oral route results in decreased transmission efficiency

PONE-D-20-23931R1

Dear Dr. Greenlee,

We’re pleased to inform you that your manuscript has been judged scientifically suitable for publication and will be formally accepted for publication once it meets all outstanding technical requirements.

Kind regards,

Byron Caughey

Academic Editor

PLOS ONE
---

## [Editor Report · Acceptance letter]

19 Nov 2020

PONE-D-20-23931R1 

Autoclave treatment of the classical scrapie agent US No. 13-7 and experimental inoculation to susceptible VRQ/ARQ sheep via the oral route results in decreased transmission efficiency 

Dear Dr. Greenlee:

I'm pleased to inform you that your manuscript has been deemed suitable for publication in PLOS ONE. Congratulations! Your manuscript is now with our production department. 

Kind regards, 

on behalf of

Dr. Byron Caughey 

Academic Editor

PLOS ONE